# Broad Spectral Response FeOOH/BiO_2−x_ Photocatalyst with Efficient Charge Transfer for Enhanced Photo-Fenton Synergistic Catalytic Activity

**DOI:** 10.3390/molecules29040919

**Published:** 2024-02-19

**Authors:** Pengfei Wu, Yufei Qin, Mengyuan Gao, Rui Zheng, Yixin Zhang, Xinli Li, Zhaolong Liu, Yingkun Zhang, Zhen Cao, Qingling Liu

**Affiliations:** 1Tianjin Key Laboratory of Indoor Air Environmental Quality Control, School of Environmental Science and Engineering, Tianjin University, Tianjin 300350, China; wupengfei1108@163.com; 2Hebei Pollution Control Technology Innovation Center of Steel and Coking Industry, Department of Environmental and Chemical Engineering, Hebei Vocational University of Industry and Technology, Shijiazhuang 050091, China; Q180311298934@163.com (Y.Q.); zhengrui0511@126.com (R.Z.); hbzhangyixin128@163.com (Y.Z.); lixl18631113353@sina.com (X.L.); jl03230619@gmail.com (Z.L.); 3Hebei Provincial Academy of Ecological Environmental Science, Shijiazhuang 050030, China; 15127162988@163.com; 4Hebei Key Lab of Environmental Photocatalytic and Electrocatalytic Materials, College of Chemical Engineering, North China University of Science and Technology, Tangshan 063210, China; 15614189119@163.com

**Keywords:** FeOOH, BiO_2−x_, hydrogen peroxide, wide spectral response, photo-Fenton

## Abstract

In this work, to promote the separation of photogenerated carriers, prevent the catalyst from photo-corrosion, and improve the photo-Fenton synergistic degradation of organic pollutants, the coating structure of FeOOH/BiO_2−x_ rich in oxygen vacancies was successfully synthesized by a facile and environmentally friendly two-step process of hydrothermal and chemical deposition. Through a series of degradation activity tests of synthesized materials under different conditions, it was found that FeOOH/BiO_2−x_ demonstrated outstanding organic pollutant degradation activity under visible and near-infrared light when hydrogen peroxide was added. After 90 min of reaction under photo-Fenton conditions, the degradation rate of Methylene Blue by FeOOH/BiO_2−x_ was 87.4%, significantly higher than the degradation efficiency under photocatalysis (60.3%) and Fenton (49.0%) conditions. The apparent rate constants of FeOOH/BiO_2−x_ under photo-Fenton conditions were 2.33 times and 3.32 times higher than photocatalysis and Fenton catalysis, respectively. The amorphous FeOOH was tightly coated on the layered BiO_2−x_, which significantly increased the specific surface area and the number of active sites of the composites, and facilitated the improvement of the separation efficiency of the photogenerated carriers and the prevention of photo-corrosion of BiO_2−x_. The analysis of the mechanism of photo-Fenton synergistic degradation clarified that ·OH, h^+^, and ·O_2_^−^ are the main active substances involved in the degradation of pollutants. The optimal degradation conditions were the addition of the FeOOH/BiO_2−x_ composite catalyst loaded with 20% Fe at a concentration of 0.5 g/L, the addition of hydrogen peroxide at a concentration of 8 mM, and an initial pH of 4. This outstanding catalytic system offers a fresh approach to the creation and processing of iron-based photo-Fenton catalysts by quickly and efficiently degrading various organic contaminants.

## 1. Introduction

With the rapid advancement of industrialization and urbanization, severe environmental pollution has become a global concern, evolving into an increasingly grave public health issue. Disruptive organic pollutants, including phenols, antibiotics, pesticides, organic dyes, pharmaceuticals, and personal care products, etc., are discharged into the environment from sectors such as pharmaceuticals, paper production, printing, petrochemicals, and daily chemical usage. These pollutants, in particular, contaminate and endanger aquatic ecosystems. However, it is challenging to remove these contaminants using existing chemical and biological water treatment techniques [1,2]. Therefore, finding new and efficient methods for wastewater treatment has become particularly crucial. Photocatalytic technology has gained widespread use in various applications [3,4,5]. One of the main benefits is that light-driven methods have higher energy efficiency as they use solar energy, which is a renewable energy source, to start reactions at lower energy thresholds and therefore use less energy overall [6]. However, current photocatalysts often face challenges such as a poor response to visible light and susceptibility to photogenerated carrier recombination, restricting the production of active oxide species [7]. The Fenton reaction has also garnered significant attention from researchers as an efficient advanced oxidation technique. It can non-selectively degrade organic pollutants using ·OH radicals generated during the reaction. Nevertheless, the homogeneous Fenton reaction has inherent limitations, including a narrow applicable pH range, the requirement for a large amount of H_2_O_2_ addition, the generation of substantial iron sludge, and a low degree of mineralization of organic pollutants [8,9,10,11,12]. These limitations constrain its practical applications.

In recent years, bismuth-based semiconductor photocatalytic materials have garnered extensive attention due to their broad photoresponse range and easily tunable electronic and morphological properties. The mutual overlap of their O 2p and Bi 6s orbitals can effectively reduce the band gap and widen the valence band, creating a unique electronic structure that enhances electronic conductivity and facilitates efficient charge separation. This distinctive electronic configuration contributes significantly to the enhancement of photocatalytic performance [13,14,15]. Various trivalent or pentavalent bismuth-based semiconductor photocatalytic materials, including Bi_2_O_3_ [16], Bi_2_MoO_6_ [17,18], Bi_2_WO_6_ [19,20], BiVO_4_ [21], NaBiO_3_ [22], and BiOX (X = Cl, Br, I) [23,24,25], have found widespread application in the photocatalytic removal of organic pollutants. Notably, BiO_2−x_ is a narrow bandgap (<2 eV) semiconductor exhibiting excellent visible light responsiveness (up to 850 nm) [26]. Li et al., in their research, pointed out that BiO_2−x_ offers a shorter path for the diffusion of photogenerated carriers due to its two-dimensional layered structure with abundant oxygen vacancies and a large number of uncoordinated surface atoms. Simultaneously, this structure exposes more active sites, thus hindering the recombination of electron–hole pairs [27]. Liu et al. proposed that the photogenerated electrons excited from BiO_2−x_, as a semiconductor material with mixed valence states, can jump between Bi^3+^ and Bi^5+^ to achieve a narrower forbidden bandwidth and extend its optical response range [28].

The non-stoichiometric BiO_2−x_ photocatalysts with zero-dimensional point defects (vacancies and substitution) and tunability are crucial to the catalytic process [29]. These defects profoundly enhance catalytic performance by modulating the electronic structure, light absorption properties, and surface active sites of the materials [30,31,32]. Vacancies and substitution defects, for instance, induce localized electric fields within the crystal, effectively separating electrons and holes and limiting their recombination during migration from the crystal interior to the surface [33,34]. Additionally, surface oxygen vacancies offer coordination unsaturation sites, promoting strong interactions with molecular oxygen, ensuring chemisorption, and facilitating electron transfer. This synergistic interaction of double defects significantly amplifies catalytic performance [35,36]. However, these unique features of BiO_2−x_ are often accompanied by some shortcomings. While BiO_2−x_ with a single point deficiency may offer active sites for improved interaction with oxygen molecules to produce reactive oxygen species, it may also diminish the use of photogenerated carriers and decrease the photocatalytic effectiveness [37]. Moreover, BiO_2−x_ photocatalysts are plagued by photo-corrosion, where photogenerated electrons can reduce Bi^5+^ to Bi^3+^, leading to a decline in catalytic performance [38,39,40].

To address these challenges, the acceleration of photogenerated electron transfer can be achieved through the construction of heterojunctions. In a previous study, FeOOH was identified as an environmentally friendly, cost-effective, and readily available adsorbent for the removal of heavy metal ions or organic pollutants from water bodies [41,42]. Researchers Zhao et al. highlighted that FeOOH not only adsorbs organic dyes such as Methylene Blue (MB) but also the Fe(III)/Fe(II)/H_2_O_2_ catalytic system exhibits a high rate of organic decomposition [43]. Consequently, we propose that the inherent weak redox capacity of BiO_2−x_ can be effectively addressed by forming a composite with FeOOH and optimizing the Fe(III)/Fe(II) ratio. Simultaneously, the utilization of high and low valence state transitions of Fe ions as a bridge for the photocatalytic-Fenton reaction can significantly enhance ·OH generation. This approach not only mitigates the issue of facile recombination of photogenerated electron–hole pairs but also enhances the rate of the Fenton reaction, mitigates iron sludge formation, and manifests remarkable synergistic effects.

Taking into account the considerations mentioned above, we encapsulated BiO_2−x_ with amorphous FeOOH to prepare a binary composite photocatalyst. This approach was undertaken to mitigate the photocorrosion of BiO_2−x_ and assess the effectiveness of this composite for the degradation of organic pollutants within the photo-Fenton system. This study delves into the synthesis process of FeOOH/BiO_2−x_, meticulously analyzing its structural, morphological, and optical attributes. Moreover, we investigate its efficacy in degrading organic substances under various conditions and explore potential degradation mechanisms. Through this investigation, we address the following key points: (1) the impact of catalyst dosage, H_2_O_2_ concentration, initial pH, and catalyst concentration on the photocatalytic efficiency; (2) exploration of the primary active species within FeOOH/BiO_2−x_ composites in the context of the photo-Fenton system; (3) elucidation of the reaction mechanism of FeOOH/BiO_2−x_ within the photo-Fenton system. By addressing these complexities, our study presents a novel approach for fabricating efficient and eco-friendly bismuth-based composite catalysts, thus contributing significantly to the field of photocatalytic water pollution treatment.

## 2. Results and Discussion

### 2.1. Characterization of As-Prepared Materials

The composition and phase structure of the sample were analyzed using X-ray diffraction (XRD), as shown in Figure 1a. The XRD pattern of pure BO exhibited characteristic peaks at 28.21°, 32.69°, 46.92°, 55.64°, and 75.64°, corresponding to crystal planes (111), (200), (220), (311), and (331) respectively [44]. These peaks displayed sharp patterns and high intensity. Upon comparison with the JCPDS NO.47-1057 standard card, it was confirmed that these substances possessed a high degree of crystallinity and exceptional purity [38]. In contrast, the synthesized FeOOH did not display distinct characteristic peaks, indicating its amorphous structure. With an increasing loading ratio of FeOOH, efficient encapsulation of BO by FeOOH occurred. The amorphous FeOOH shielded the crystal structure characteristic peaks of FBO, leading to a decrease in their intensity [45]. Notably, although the intensity of the crystal characteristic peaks diminished, no new diffraction peaks appeared in any of the samples depicted in Figure 1a. This absence indicates that no additional substances were generated in the resulting products.

FTIR was employed to analyze the structure and chemical composition of various samples, as shown in Figure 1b. In pure BO, two strong characteristic bands emerge near 590 cm^−1^ and 530 cm^−1^, representing the stretching vibrations of the Bi-O bond in the BiO_3_ pyramidal structural unit and BiO_6_ octahedral structural unit, respectively [46]. For pure FeOOH, bend vibration absorption bands appear at ~890 cm^−1^ and ~790 cm^−1^, caused by the in-plane bending of groups on the surface of Fe-OH-Fe, while the stretch vibration absorption band near 630 cm^−1^ corresponds to the Fe-O bond [47]. The characteristic absorption bands of the xFBO (x = 5, 10, 20) closely resemble those of pure BO, owing to the small amount of added FeOOH. Notably, at a mass percentage of 10% FeOOH, the characteristic absorption band exhibits the highest intensity.

SEM, TEM, and EDS techniques were employed to investigate the microscopic morphological characteristics and the distribution of elements of BO and 20FBO. In Figure 2a,c, SEM and TEM images of the BO monomer are presented, revealing its irregularly stacked layered structure. Figure 2d displays a TEM photograph of the thin layer on the edge of the BO monomer, where orderly lattice stripes are clearly visible with lattice spacings of 0.317 nm and 0.274 nm, corresponding to the (111) and (200) crystal planes of BiO_2−x_, respectively, confirming its crystal structure [48]. Figure 2b shows the SEM image of the composite 20FBO. It can be seen that FeOOH grows on the surface of BiO_2−x_, forming a three-dimensional flocculent microsphere. Further detailed structure analysis was conducted using TEM, revealing that the BiO_2−x_ surface is uniformly coated by flocculated FeOOH, as depicted in Figure 2e. To confirm the elemental distribution of the composite, EDS analysis was performed. The results, shown in Figure 2f, indicate the presence of three elements: Fe, Bi, and O. Fe is uniformly distributed on the flakes of BiO_2−x_, confirming the effective coating of FeOOH on BiO_2−x_.

The specific surface area and pore structure of the prepared catalysts were analyzed using nitrogen adsorption–desorption technology. Figure 3a,b illustrates the nitrogen adsorption–desorption isothermal curves of BO and 20FBO samples. According to the IUPAC classification, both BO and 20FBO exhibit typical type IV curves, while the 20FBO displays a pronounced H3-type hysteresis loop over a wider range of relative pressures (P/P_0_ = 0.2–1.0). This indicates mesoporous characteristics generated by the flocculent FeOOH particles anchored to the surface of the BO [49]. Compared to BO, 20FBO has a wider pore size distribution, a larger hysteresis loop, and a larger specific surface area, indicating more and mostly conical mesopores. The results showed that the specific surface area of pure BO was 11.023 m^2^/g, whereas the specific surface area of 20FBO loaded with FeOOH was significantly increased to 72.952 m^2^/g, which was an increase of about 6.6 times. The addition of FeOOH increases the mesopore proportion and pore volume, enhancing the adsorption of organic pollutants and providing more catalytically active sites, improving reaction speed and selectivity in catalytic reactions.

The optical absorption of pure BO, and different ratios of FBO composites were analyzed based on the UV–Vis DRS spectrum, as depicted in Figure 4a. It is evident that the light-trapping ability of the FBO composites is enhanced in the UV, Visible, and NIR regions. With the increase in FeOOH loading, the light absorption band edges of the FBO composites were also slightly red-shifted. Specifically, the absorption band edges of 20FBO expanded from 800 nm to 834 nm. This augmented light absorption capacity stimulates increased charge carrier production, leading to enhanced sunlight utilization and improved photocatalytic performance of the catalysts. Notably, the color of the catalyst changes from dark brown to brownish yellow with the increased presence of FeOOH.

The band gap energies of the obtained photocatalysts were calculated using the transformed Kubelka–Munk function (Equation (1)).
*αhυ = A*(*hυ* − *E_g_*)*^n^*(1)

The absorption coefficient (*α*), optical frequency (*υ*), proportionality constant (*A*), bandgap energy (*E_g_*), and the transition types of the semiconductor samples (*n*) are represented in the equation, where FBO is considered an indirect semiconductor (*n* = 2). By extrapolating the lines to the *hυ* axis, the band gap energies of BO and 20FBO are determined as 1.85 eV and 1.79 eV, respectively (Figure 4b). The narrower band gap facilitates electron excitation, promoting effective separation and rapid transfer of electrons and holes, thereby enhancing photocatalytic degradation activity. To study their band structures, the valence band XPS spectra were measured, and the results are shown in Figure 4c. The valence band (VB) potentials of BO and 20FBO are evaluated at 0.71 eV and 0.94 eV, respectively. The BO and 20FBO conduction bands were calculated to be −1.14 eV and −0.85 eV, according to the formula E_CB_ = E_VB_ − *E_g_* [50].

### 2.2. Evaluation of Photocatalytic Performance

The degradation activities of BO and 20FBO were studied to compare their efficiency in degrading MB under photocatalytic, Fenton, and photo-Fenton conditions. According to the results in Figure 5a, BO has low degradation activity in all three degradation conditions. It is noteworthy that the degradation rate was 33.4% at 90 min in the BO-Vis system, which instead decreased to 25.1% at 90 min in the BO-Vis-H_2_O_2_ system. This result may be attributed to the abundant oxygen vacancies on the surface of the BO lamellae, which serve as active sites for molecular oxygen adsorption and activation, generating active oxygen species involved in the degradation reaction. However, the introduction of H_2_O_2_ to the system reduced the number of oxygen vacancies on the BO surface. This reduction weakened the adsorption effect, leading to the decrease in the degradation rate [51,52]. However, upon successful modification with FeOOH, the catalytic activity of the 20FBO composite significantly increased. Under photo-Fenton conditions for 90 min, 20FBO efficiently degraded MB at 87.4%, surpassing the degradation efficiencies of photocatalysis (60.3%) and Fenton (49.0%). From the data results, it can be seen that the composites do have a stronger adsorption capacity for the pollutants, which promotes the catalytic performance of the catalysts. However, absorption is not the main reason for pollutant degradation. This enhancement is attributed to the increased specific surface area and mesopore ratio due to FeOOH loading, providing more active sites for organic pollutant adsorption. Furthermore, the FeOOH-modified composite demonstrated higher electron-hole separation efficiency and photogenerated carrier transfer efficiency, accelerating the photocatalytic reaction. The addition of H_2_O_2_ played a vital role in the photo-Fenton reaction, promoting Fe^3+^/Fe^2+^ cycling and enhancing the efficiency of ·OH generation, further enhancing photocatalytic degradation of MB.

To assess the photocatalytic capability of the samples, experimental data were fitted using the apparent pseudo-first-order model (ln(*C_0_*/*C*) = *kt*), as shown in Figure 5b. The corresponding slope of the fitted line represents the apparent rate constant (*k*) of MB degradation. Under photo-Fenton conditions, the *k* values of 20FBO are 2.33 times and 3.32 times higher than photocatalysis and Fenton catalysis, respectively, highlighting the significant role of the photo-Fenton synergistic effect in MB degradation. In Figure 5c, the 2D UV–Visible absorption spectrum of MB degradation by 20FBO under photo-Fenton conditions was depicted. As the reaction progresses, the intensity of the MB absorption bands gradually diminishes. After 90 min, the absorption peak dwindles to a minimal level, indicating that the vast majority of MB had been degraded.

Meanwhile, a xenon lamp with a 780 nm cut-off filter was employed to simulate the NIR light source, aiming to verify the photocatalytic activity of the prepared materials in a broader spectrum. Analyzing the experimental data presented in Figure 5d, it is evident that both BO and 20FBO exhibit photocatalytic activity. However, it is worth noting that the energy of NIR light is comparatively lower, resulting in a reduced degradation rate observed at 90 min. This observation aligns with the findings illustrated in Figure 4. Through a literature review, a comparison of the degradation performance of MB by some different photocatalysts was carried out and can be found in the Appendix A, as shown in Appendix A.

### 2.3. Photo-Fenton Synergy Mechanism

To investigate the remarkable degradation of Methylene Blue by 20FBO under the photo-Fenton system, the chemical compositions and electronic states of the surface elements of BO and 20FBO before and after the reaction were analyzed using X-ray photoelectron spectroscopy (XPS). Appendix A displays the full XPS spectra of the samples before and after the reaction, revealing characteristic peak signals of Bi, O, Fe, and C. Interestingly, there is no noticeable change in these signals before and after the reaction, indicating the absence of new substance formation during the process.

The high-resolution Bi 4f XPS spectra of BO and 20FBO both before and after the reaction are displayed in Figure 6a,b. Fresh BO has two binding energy peaks at 163.8 eV (Bi 4f 5/2) and 158.5 eV (Bi 4f 7/2), respectively, where the peaks at 164.0 eV and 158.7 eV belong to Bi^5+^, and the peaks at 163.6 eV and 158.2 eV belong to Bi^3+^, which corresponds to the mixed valence state of BiO_2−x_ (Figure 6a) [53]. The 20FBO binding energy peaks were 164.2 eV and 158.9 eV, respectively, showing a 0.4 eV shift towards higher values compared to BO. This shift enhanced the electronegativity of Bi, indicating electronic interaction between FeOOH and BiO_2−x_ (Figure 6b). The Bi^3+^ content was increased after the BO reaction, which may be due to the photocorrosion of BO, resulting in the reduction of Bi^5+^ to Bi^3+^, whereas the proportion of Bi^3+^ did not change significantly in BiO_2−x_ modified by FeOOH. This indicates that FeOOH modification mitigates the photocorrosion of BiO_2−x_, acting as a protective layer [54].

In Figure 6c,d, the O 1s characteristic peaks can be observed, which are fitted by three peaks corresponding to lattice oxygen, oxygen vacancies, and chemisorbed oxygen [55]. Upon loading FeOOH, the O 1s characteristic peak shifted from 529.3 eV to 529.8 eV, indicating a binding energy increase of 0.5 eV. This shift suggests that central metal electrons on the surface of BiO_2−x_ transferred to FeOOH as electron acceptors. Oxygen vacancies are detectable in the O 1s high-resolution spectra of both BO and 20FBO. However, there is a noticeable reduction in the oxygen vacancy content in BO as the reaction progresses. In contrast, the decrease in 20FBO oxygen vacancies is less prominent. Oxygen vacancies contain local electrons, accelerating H_2_O_2_ decomposition, promoting ·OH formation, and inhibiting photogenerated electron–hole recombination. These factors contribute to the high photocatalytic activity of the composites, aligning with the experimental results mentioned above.

The primary peaks of Fe 2p 3/2 and Fe 2p 1/2 in 20FBO before and after the reaction appeared at 711.2 eV and 724.5 eV, respectively, aligning with the characteristic values of Fe^3+^ in FeOOH, as shown in Appendix A [56]. Additionally, two oscillating satellite peaks emerge at 717.5 eV and 728.8 eV, indicating charge transfer or oscillating processes associated with Fe. Notably, after the photocatalyst reaction, the Fe 2p 3/2 and Fe 2p 1/2 peaks shift by 0.3 eV and 0.4 eV towards reduced binding energy. This shift is primarily attributed to the capture of some photogenerated electrons by Fe, leading to a decrease in electronegativity.

Figure 7a illustrates the transient photocurrent responses of BO, 20FBO, and 20FBO-H_2_O_2_. It is evident that all the samples exhibit rapid responses to light irradiation, with the maximum transient photocurrent response repeating within eight on/off photoperiods. Compared to BO, 20FBO displays a stronger photocurrent response, indicating that the modification of FeOOH enhances the efficiency of photogenerated electron–hole separation, which leads to an increase in the number of photogenerated carriers and their transport capacity. It is worth noting that the pure BO photocurrent density tends to decline over time due to photocorrosion. In contrast, the photogenerated electrons in the composites, after the coating of FeOOH, are quickly transferred to FeOOH, protecting BiO_2−x_ from photocorrosion. Thus, the photocurrent density remains stable over time. Upon the addition of H_2_O_2_, there is a significant decrease in the photocurrent density. This decrease is attributed to the involvement of electrons transferred to FeOOH in the Fenton reaction [57]. Consequently, the electrons cannot pass through the external test circuit. This demonstrates the synergistic effect of photocatalysis and Fenton, leading to rapid charges transfer and improved photocatalytic activity [58]. The charge transfer efficiencies of BO and 20FBO were investigated using electrochemical impedance spectra (EIS), and the results are depicted in Figure 7b. The arc radius of 20FBO is significantly smaller than that of BO, indicating that its photogenerated charge experiences less resistance during transportation between interfaces. This property facilitates the efficient separation of photogenerated electron–hole pairs, enhancing the photocatalytic performance of the composites.

The electron spin resonance (ESR) technique was employed to investigate the primary active species of 20FBO composites during the degradation of MB in the photo-Fenton system, and DMPO was used to capture the active radical species produced during the photodegradation. The dispersant used for hydroxyl radicals is water, while the dispersant used for superoxide radical detection is methanol, as shown in Figure 8a,b. In the absence of light, the ESR signals of DMPO-·O_2_^−^ were undetectable, and weak DMPO-·OH signals were observed due to the limited ·OH production resulting from the reaction between 20FBO and H_2_O_2_ [59]. Under illuminated conditions, it was clearly observed that DMPO-·O_2_^−^ corresponded to the 1:1:1:1:1 four-signal characteristic peaks and DMPO-·OH corresponded to the 1:2:2:1 four-signal characteristic peaks [60]. With the passage of time, both feature peaks intensities increased. This observation indicates that 20FBO reacts with H_2_O_2_ to generate ·O_2_^−^ and ·OH under light conditions.

In order to assess the significance of different active species in the degradation process, various quenchers were added, and quenching experiments were conducted (Figure 8c). Isopropyl alcohol (IPA) and triethanolamine (TEOA) were utilized as quenchers to inhibit ·OH and h^+^, respectively [27]. The results indicated that the degradation process was notably hindered by the addition of IPA, resulting in a degradation rate of only 41.7% at 90 min. Conversely, the inhibitory effect of adding TEOA was not significant, leading to a degradation rate decrease to 79.3%. To investigate the influence of dissolved oxygen on ·O_2_^−^ formation, nitrogen was introduced. Nitrogen purging decreased the dissolved oxygen content, inhibiting ·O_2_^−^ formation on the catalyst surface and slightly decreasing degradation efficiency. Conversely, the introduction of oxygen increased ·O_2_^−^ production, resulting in a slight increase in degradation efficiency [61]. These findings suggest that ·OH is the primary active substance involved in the MB degradation process, while h^+^ and ·O_2_^−^ serve as secondary active substances with relatively minor contributions.

To elucidate the mechanism of ·OH radical generation under photo-Fenton synergism, we employed the fluorescent coumarin assay to detect ·OH fluorescence. Coumarin, serving as a fluorescent probe, reacts with ·OH to yield hydroxycoumarin, resulting in a distinct fluorescence peak around 450 nm [62]. Figure 9a illustrates that the fluorescence peaks intensified over time, indicating continuous ·OH generation within the reaction system. The faint fluorescence peaks observed at 0 min could be attributed to ·OH generated by the Fenton reaction between Fe^2+^ and H_2_O_2_ in the system or the activation of oxygen vacancies in the catalytic system. This activation facilitates H_2_O_2_ decomposition, producing ·OH radicals, aligning with the experimental findings in DMPO-·OH spin-trapping ESR spectra, as shown in Figure 8b.

The decomposition of H_2_O_2_ at different reaction times under the photo-Fenton system is depicted in Figure 9b. A yellow complex forms between H_2_O_2_ and TiOSO_4_/H_2_SO_4_, enabling the measurement of H_2_O_2_ consumption via UV–visible spectrophotometry during the reaction [63]. The H_2_O_2_ content gradually decreases as the reaction progresses, indicating continuous decomposition of H_2_O_2_ into ·OH under the photo-Fenton conditions. Combined with the results of the above catalyst degradation activity experiments (Figure 5a), the degradation activity of 20FBO without the addition of H_2_O_2_ is much lower than that under photo-Fenton conditions, which suggests that the source of ·OH mainly relies on the decomposition of H_2_O_2_ to generate, while the generation of ·OH by photocatalysis alone is limited.

Based on the preceding results and discussions, a mechanism for the photo-Fenton synergistic degradation of MB by the 20FBO composite was proposed, as illustrated in Figure 10. Due to the existence of a large number of oxygen vacancies on the surface of BiO_2−x_ as the electron trapping centers, and the interaction between BiO_2−x_ and FeOOH to form a ligand bond that facilitates electron transfer to Fe, the electron–hole pairs generated by visible light irradiation can be separated quickly, which improves the effective number of electrons. Part of the electrons adsorbed on the surface of the catalyst react with oxygen molecules to generate ·O_2_^−^, and the remaining part of the electrons react with FeOOH in the composite material to reduce Fe^3+^ to Fe^2+^, which not only inhibits the photocorrosion effect of BiO_2−x_, but also triggers a Fenton reaction under the participation of H_2_O_2_ to rapidly generate ·OH. Due to the existence of the oxygen vacancies on the surface of the composite material, the O-O bond in the H_2_O_2_ is adsorbed and weakened by stretching [64] and 3d orbitals of Fe^2+^ and sp3 hybridization orbitals of O-O bonds partially overlap, further weakening the O-O bond [58], so that Fe^2+^ is more likely to be weakened in the reaction system of H_2_O_2_ to generate ·OH, Fe^2+^ and then oxidized into Fe^3+^, forming the Fe^3+^/Fe^2+^ cycle, which not only solves the problem of secondary pollution in the traditional Fenton water treatment process but also improves the efficiency of degradation of pollutants and fulfills the synergistic effect of photo-Fenton [56].

### 2.4. Optimization of Degradation Reaction Conditions

The photo-Fenton degradation efficiency of MB was significantly enhanced by introducing FeOOH onto BiO_2−x_. Subsequently, the impact of various test parameters on MB degradation efficiency over 90 min was investigated. In Figure 11a, the degradation efficiencies of FBO with different FeOOH loading ratios are presented. The degradation efficiencies for 5FBO, 10FBO, 20FBO, 30FBO, and 40FBO were 62.1%, 84.9%, 90.9%, 75.5%, and 52.9%, respectively. The MB degradation efficiency progressively increased with the higher proportion of FeOOH. Notably, 20FBO exhibited the highest catalytic activity, while continued addition led to a decrease in degradation efficiency. This may be due to the fact that when the amount of FeOOH attached to the BiO_2−x_ surface decreases, the specific surface area of the composite decreases and consequently, the amount of pollutant adsorbed decreases as well as the number of available active sites decreases. The lower Fe content involved in the photo-Fenton synergistic cycle affects the rate of ·OH generation, inhibiting the progress of the photo-Fenton reaction. With increased FeOOH loading on BiO_2−x_, FeOOH acts as an electron acceptor, efficiently transporting photogenerated carriers to the metal active center. This process completes the separation of electron–hole pairs, leading to enhanced photocatalytic performance. However, with the increase in FeOOH loading, agglomeration occurred on the surface of BiO_2−x_ reducing the number of active sites as well as affecting the light utilization, which ultimately reduced the photocatalytic activity [58].

Figure 11b illustrates the impact of various initial pH values on the catalytic performance of the 20FBO photo-Fenton system. The initial pH was set above 4 to eliminate the influence of the homogeneous Fenton system, ensuring that MB degradation predominantly occurred on the catalyst surface. The results reveal that the 20FBO photo-Fenton system exhibits optimal degradation efficiency at an initial pH of 4~5, achieving an MB degradation rate of 97.4%. The catalytic effectiveness gradually diminishes with increasing initial pH, reaching a decomposition rate of only 51.4% at pH = 11. This suggests that the catalyst demonstrates heightened degradation activity under acidic or neutral conditions. This phenomenon may be attributed to the adsorption of numerous OH^–^ groups, displacing H_2_O_2_ on the catalyst surface in alkaline conditions, affecting the generation of ·OH. Moreover, alkaline conditions are more likely to induce the reaction of Fe^3+^ with OH^−^, forming precipitates deposited on the catalyst surface. This is unfavorable for the cycling of Fe(III)/Fe(II), resulting in decreased catalyst activity [65]. The catalyst displays robust catalytic activity under both acidic and neutral conditions, indicating a broad pH adaptation range. This characteristic contributes to significant cost savings in practical applications by eliminating the need for extensive pH adjustments.

Figure 11c illustrates the comparison of degradation efficiency at various catalyst concentrations. The results indicate that at a catalyst concentration of 0.25 g/L, the degradation rate of MB reached only 75.5%. As the catalyst concentration increased from 0.25 g/L to 1 g/L, the degradation rate rose to 91.5%. However, with a continued increase in catalyst concentration, degradation efficiency gradually decreased. This phenomenon can be attributed to the increased catalyst addition, providing more active and adsorption sites. This makes pollutants more susceptible to adsorption and reaction, generating more electrons to facilitate the conversion of Fe(III) to Fe(II). This process accelerates the decomposition of H_2_O_2_, enhancing catalytic performance. When the catalyst concentration surpasses the optimal dosage, particles agglomerate significantly, negatively impacting the normal progression of the catalytic reaction. Excessive catalyst aggregation leads to increased opacity and light scattering, reducing the catalyst’s light capture ability. This effect seriously hampers electron excitation and diminishes the number of ·OH radicals, thereby decreasing the activity of the photo-Fenton system. An excess of catalyst in the reaction system also results in surplus Fe^2+^, which consumes ·OH in the system, consequently reducing the removal rate of MB [62].

From Figure 11d, it can be observed that as the H_2_O_2_ concentration in the system increases from 4 mM to 8 mM, the corresponding degradation rate rises from 79.9% to 90.9%. This is attributed to the beneficial effect of an appropriate amount of H_2_O_2_ in activating the decomposition into ·OH. However, as the H_2_O_2_ concentration continues to increase to 20 mM, the degradation rate decreases to 73.3%. This decline may be attributed to the excess H_2_O_2_ acting as a scavenger for ·OH, forming ·HO_2_ (in Equation (2)). ·HO_2_ exhibits a weaker oxidation capability and further reacts with ·OH to generate O_2_ and H_2_O (in Equation (3)). This leads to the consumption of ·OH and H_2_O_2_, slowing down the reaction rate [66].
H_2_O_2_ + ·OH → H_2_O+ ·HO_2_(2)
·HO_2_ +·OH → O_2_ + H_2_O(3)

### 2.5. Degradation Test and Stability Test of Different Pollutants

A comprehensive assessment was conducted on the 20FBO composite catalyst, exploring its efficacy under optimal conditions for photo-Fenton degradation of various target pollutants, namely Methyl Orange (MO), Rhodamine B (RhB), Phenol, and Bisphenol A (BPA). The results, depicted in Figure 12a, reveal degradation rates after 90 min in the photo-Fenton system and the degradation rates of MO, RhB, Phenol, and BPA reached 93.5%, 80.3%, 73.5%, and 84.4% respectively. This implies the material is efficient in breaking down different types of contaminants. Simultaneously, mineralization rates of the five organic pollutants after 90 min of degradation were analyzed. MB, MO, RhB, Phenol, and BPA exhibited mineralization rates of 51.3%, 46.4%, 41.7%, 35.2%, and 43.8%, respectively (Figure 12b). Notably, the degree of mineralization progressively increased with prolonged reaction time, and maintaining a higher H_2_O_2_ level contributed to enhanced mineralization (Appendix A). In comparison, positively charged MB exhibited higher affinity for adsorption on the 20FBO surface, actively participating in oxidation reactions (Appendix A). Ion chromatography confirmed the presence of carbonate ions, sulfate, nitrate, and ammonium salts resulting from the mineralization of C, S, and N functional groups [50].

To investigate the long-term stability and activity of the catalyst, the BO and 20FBO were subjected to five repeated tests under individual light conditions. Similarly, the 20FBO catalyst was tested under light-Fenton conditions five times to repeat the degradation experiment. The results, presented in Figure 13a, reveal only a slight decrease in the photocatalytic efficiency of 20FBO after five cycles of experiments. In the BO, oxygen vacancies are gradually filled by oxygen molecules and adsorbed hydroxyl oxygen during the circulation process, leading to a reduction in the catalytic activity of the catalyst [61]. Modification of the BO monomer with FeOOH induces changes in the surface structure and the number of active sites, resulting in increased oxygen vacancies and a stronger capacity for pollutant adsorption [42]. The protective effect of FeOOH mitigates the likelihood of light-induced corrosion in BO. Consequently, its catalytic activity is enhanced and remains relatively stable, consistent with the results of the transient photocurrent response. Furthermore, the dissolution of iron ions in the photo-Fenton system was analyzed using atomic absorption spectroscopy. The iron ion concentration in the reaction system was maintained between 0.025 and 0.014 mg/L, significantly below the EU standard of 2.0 mg/L. This indicates that 20FBO exhibits high stability against photochemical corrosion [43].

ESR analysis was employed for a comprehensive examination of the oxygen vacancies before and after the BO reaction. Figure 13b showed that the intensity of the characteristic peaks decreased after the BO reaction, indicating that oxygen vacancies can be filled during the reaction process. This filling results in a reduction in the redox ability of the catalyst. The consequence of this decline in redox ability is an elevated probability of recombination of photogenerated electron–hole pairs, leading to a decrease in the overall catalytic activity of the catalyst. This observation aligns seamlessly with the outcomes of the aforementioned cycle tests of BO.

At the same time, the stability of the crystal structure of 20FBO after five repeated tests and before reaction was analyzed. In the XRD comparison before and after the reaction (Appendix A), the characteristic peaks corresponding to the (111), (200), (220), and (311) crystal planes are well-maintained. Importantly, there is no discernible shift in these peaks, nor the emergence of new peaks. This result signifies that the unit cell parameters remain unaltered, and no new phases are formed. Furthermore, the XPS spectra before and after the reaction affirm that the elemental composition of the material remains constant. This evidence supports the conclusion that the successful coating of FeOOH plays a pivotal role in shielding BiO_2−x_ from photocorrosion. As a result, the catalytic activity is preserved at a high level even after multiple reuse cycles.

## 3. Experiment Section

### 3.1. Synthesis of BiO_2−x_ and FeOOH/BiO_2−x_

All reagents were of analytical grade and used directly without further purification. BiO_2−x_ was synthesized through a simple hydrothermal process. The specific preparation method was as follows: 1.8 mmol of NaBiO_3_ was added to 20 mL of 3.0 M NaOH solution. The aforementioned suspension was gradually dispersed in 20 mL of deionized water to obtain a homogeneous mixture. The mixture was vigorously stirred at room temperature for 1 h. Subsequently, the suspension was transferred to a 100 mL stainless steel autoclave lined with Teflon, heated at 180 °C for 3 h, and then allowed to cool naturally to room temperature. The obtained precipitate was separated and dried at 70 °C for 12 h, then ground to obtain BiO_2−x_ powder, which was labeled as BO. A specific amount of FeCl_3_·6H_2_O and 0.2 g of BiO_2−x_ powder was dissolved in 50 mL anhydrous ethanol and stirred thoroughly for 20 min. Subsequently, 3 mmol of NH_4_HCO_3_ was added with continuous stirring at room temperature for 8 h. The resulting precipitate was separated by centrifugation, washed three times with ultrapure water and ethanol, and then dried under a vacuum at 40 °C for 12 h. The synthesized composite material FeOOH/BiO_2−x_ was denoted as xFBO (x = 5, 10, 20, 30, 40, respectively), where x represented the molar ratio of FeOOH to BiO_2−x_.

### 3.2. Characterization of Materials

The X’Pert PROMPd X-ray diffractometer (XRD, PANalytical, The Netherlands) was used for crystal structures and phase state analysis, and the scanning angle was 5°–85°, with a scanning speed of 10°/min. The GeminiSEM300 field emission scanning electron microscope (SEM, ZEISS, Germany) was used to study the surface morphology of the catalyst; and TecnaiG2F20 field emission transmission electron microscopy (TEM, FEI, U.S.) was used to study the crystal lattice and microscopic morphology of the catalyst and scan the contained elements through the EDS energy spectrum. Specific surface area was measured by the Autosorb IQ (BET, Quantachrome, U.S.) model; the pore size and pore distribution were determined using the Barrett Joyner Halenda (BJH) model. EscaLabXi X-ray photoelectron spectrometry (XPS, ThermoFisher, U.S.) was used to determine the chemical status of the samples. The functional groups of the samples were identified by VERTEX70 Fourier transform infrared spectroscopy (FT-IR, BRUKER, Germany). The TU-1950 UV–Visible diffuse reflectance spectra (UV–Vis DRS, PERSEE, China) were used to determine the light absorption characteristics of the catalyst at a scanning wavelength of 200–950 nm. The electron spin resonance (ESR) signals of radicals spin-trapped by the spin-capture reagent 5,5′-dimethyl-1-pyrrolidine N-oxide (DMPO) were examined on an A300 BRUKER system to analyze the types of active free radicals and oxygen vacancies. The F-7000FL fluorescence emission spectra (Hitachi, Japan) were used to measure the photo-excited electron–hole recombination rate of the catalyst with an excitation wavelength of 270 nm. The surface charge of the materials was determined using a Nano ZS90 Zeta potentiometer (Malvern, UK). The CHI660E (Chenhua, China) electrochemical workstation and a three-electrode reaction cell were used to test the photocurrent and impedance under the 500 W xenon lamp and a 420 nm cut-off filter.

### 3.3. Evaluation of Catalytic ACTIVITY

The photo-Fenton synergistic degradation of difficult organic micropollutants was carried out in a multi-tube photochemical reactor (AULTT CEL-LAB500E4), which was linked to a cooling device to maintain the reaction solution temperature constant at 30 ± 2 °C. The visible light source used was a 500 W Tubular Xenon lamp with a 420 nm cutoff filter (Vis: λ > 420 nm), and the near-infrared light used was a 500 W Tubular xenon lamp with a 780 nm cutoff filter (NIR: λ > 780 nm). Visible and near-infrared light sources were measured to be 70 and 50 mW·cm^−2^, separately. To allow the adsorption–desorption equilibrium, a specific quantity of catalyst was added to 50 mL of reaction solution and agitated in the dark for 30 min. After adding a certain volume of H_2_O_2_, the xenon lamp was turned on to start the degradation process. A 3 mL sample of the reaction suspension was collected at intervals. The sample was filtered through a 0.22 μm membrane and placed in darkness. The degradation degree of organic micropollutants was detected by a UV–Vis spectrophotometer (TU-1901, PERSEE, China) or high-performance liquid chromatography (Chromaster-5430, Hitachi, Japan). Using a total organic carbon (TOC) analyzer (TOC-L, Shimadzu, Kyoto, Japan), the photo-Fenton mineralization rate of organic pollutant solutions was detected.

## 4. Conclusions

In conclusion, this work controllably synthesized a coating structure of an FeOOH/BiO_2−x_ composite with a broad spectral response and high electron transfer efficiency and constructed a photo-Fenton synergistic catalytic degradation system by adding H_2_O_2_. The developed system not only demonstrates outstanding degradation efficiency for the targeted pollutants but also showcases remarkable photochemical stability. These exceptional properties can be attributed to the following factors: (1) BiO_2−x_ exhibits a narrower forbidden bandwidth, enhancing solar energy utilization and elevating the excitation of photogenerated electrons. The interaction between BiO_2−x_ and FeOOH, leading to the formation of a ligand bond, facilitates the transfer of electrons to Fe. This separation of electron–hole pairs, generated through excitation under visible light irradiation, swiftly increases the number of effective electrons. (2) The application of the FeOOH on the surface of the material not only mitigates the photocorrosion of BiO_2−x_, ensuring catalytic stability, but also enables photogenerated electrons to reduce Fe(III) to Fe(II). Simultaneously, the Fenton reaction engenders ·OH through the co-participation of H_2_O_2_. (3) The substantial specific surface area in the amorphous state of FeOOH facilitates the adsorption of organic pollutants. Concurrently, the oxygen vacancies in BiO_2−x_ not only adsorb and activate molecular oxygen, generating ·O_2_^−^, but also catalyze the breaking of the O-O bond in H_2_O_2_, resulting in the generation of ·OH, which can increase the number of active radicals and enhance the photo-Fenton co-catalysis performance. This excellent catalytic system can rapidly and effectively degrade different types of organic pollutants, providing a new idea for the design and preparation of iron-based photo-Fenton catalysts.

## Figures and Tables

**Figure 1 molecules-29-00919-f001:**
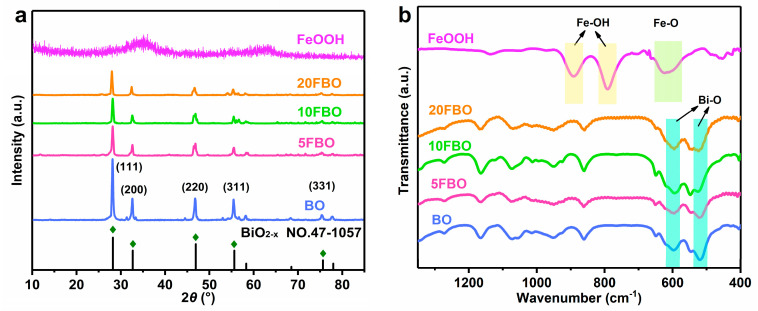
(**a**) XRD patterns and (**b**) FTIR spectra of as-prepared samples.

**Figure 2 molecules-29-00919-f002:**
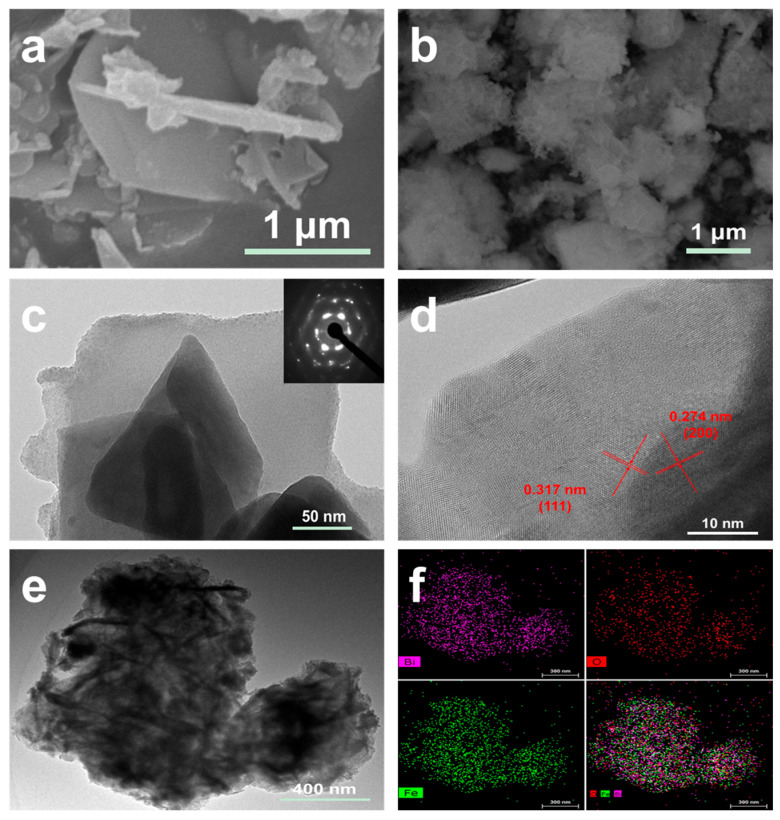
SEM image of (**a**) BO and (**b**) 20FBO; TEM image of (**c**) BO and (**e**) 20FBO; (**d**) HRTEM image of BO and (**f**) EDS mapping of 20FBO.

**Figure 3 molecules-29-00919-f003:**
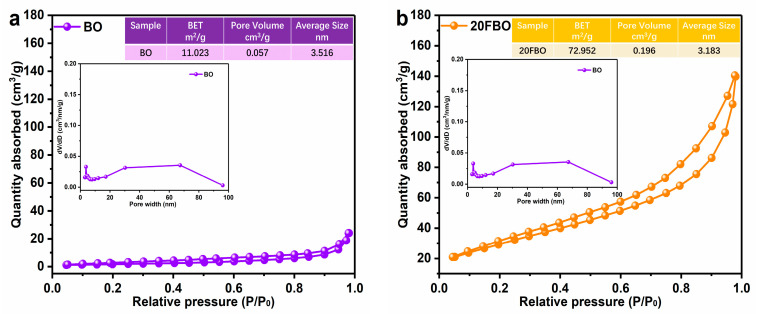
N_2_ adsorption–desorption isotherms and pore size distribution of (**a**) BO; (**b**) 20FBO composites.

**Figure 4 molecules-29-00919-f004:**
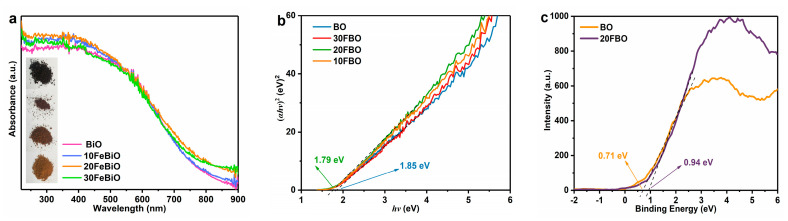
(**a**) UV−Vis diffuse reflectance spectra (UV−Vis DRS) of samples; (**b**) plots of (*αhv*)^2^ versus photon energy (*hv*) of samples; (**c**) valence band XPS spectra of samples.

**Figure 5 molecules-29-00919-f005:**
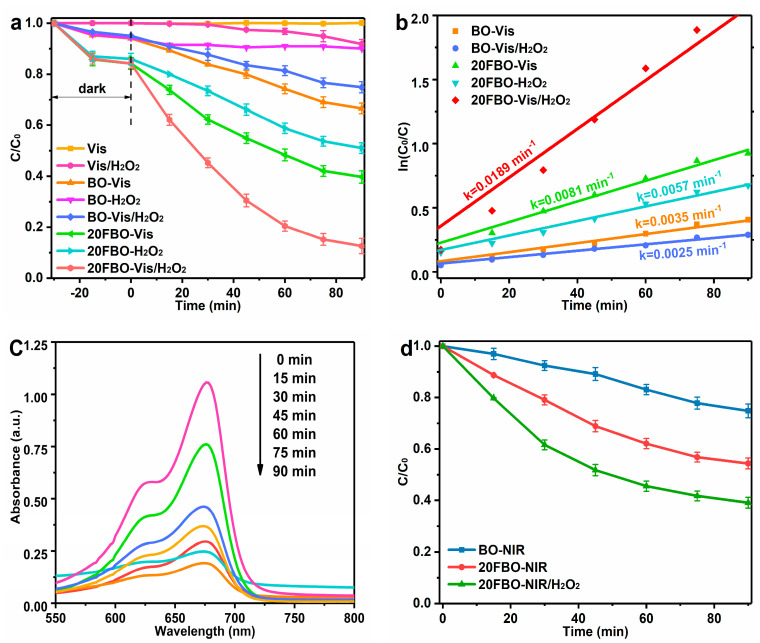
(**a**) Time profiles and (**b**) kinetic rate constants of MB degradation on the BO and 20FBO under different catalytic conditions; (**c**) 2D UV–Visible absorption spectrum for photo-Fenton MB degradation; (**d**) time profiles of MB degradation on the BO and 20FBO under NIR (conditions: MB concentration: 50 mg/L, catalyst concentration: 0.5 g/L, H_2_O_2_ concentration: 8 mM, initial pH = 7, illuminant source: 500 W Xe-lamp (Vis: λ > 420 nm; NIR: λ > 780 nm), temperature: 30 ± 2 °C).

**Figure 6 molecules-29-00919-f006:**
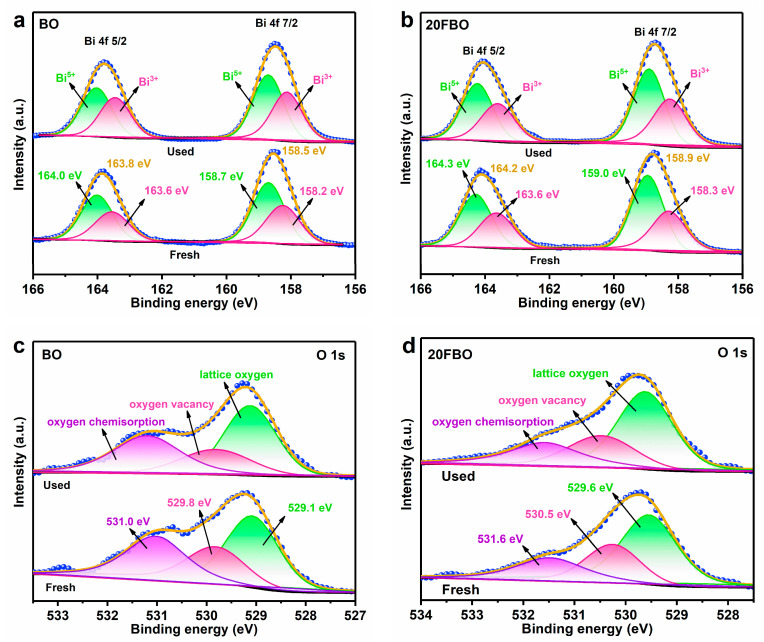
High-resolution XPS spectra of (**a**) Bi 4f and (**c**) O 1s over BO; (**b**) Bi 4f and (**d**) O 1s over 20FBO.

**Figure 7 molecules-29-00919-f007:**
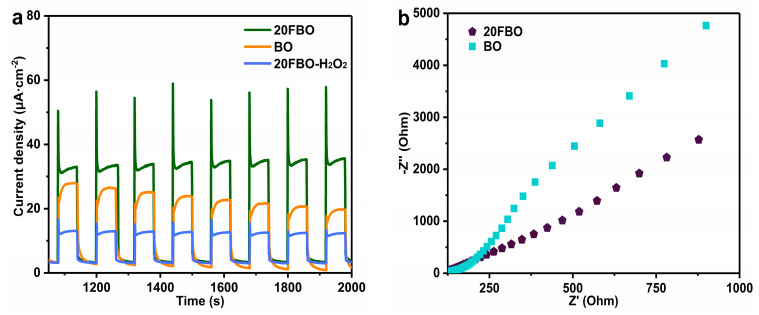
(**a**) Photocurrent spectra and (**b**) electrochemical impedance spectra under open−circuit potential conditions of pure BO and 20FBO composites.

**Figure 8 molecules-29-00919-f008:**
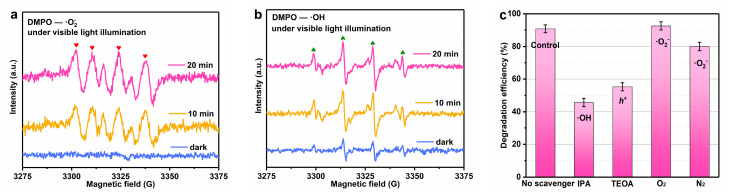
(**a**) DMPO−·O_2_^−^ spin-trapping ESR spectra in methyl alcohol in the presence of 20FBO with H_2_O_2_; (**b**) DMPO−·OH spin-trapping ESR spectra in water in the presence of 20FBO with H_2_O_2_; (**c**) comparison of photo-Fenton MB degradation in the presence of various quenchers.

**Figure 9 molecules-29-00919-f009:**
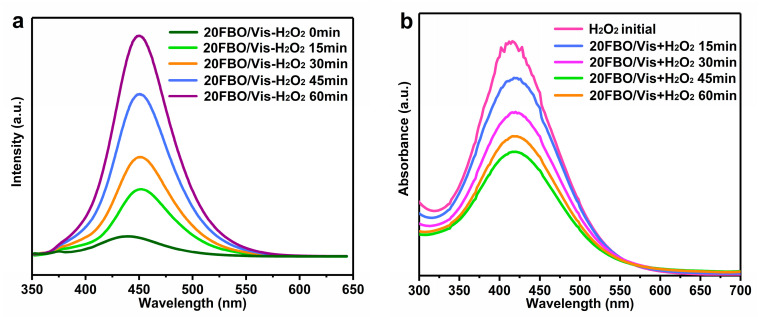
(**a**) ·OH of fluorescence spectra over time (1 mM coumarin, 0.5 g/L cat., 8 mM H_2_O_2_); (**b**) UV–Vis spectrum of H_2_O_2_-TiOSO_4_ over time (0.5 g/L cat., 8 mM H_2_O_2_).

**Figure 10 molecules-29-00919-f010:**
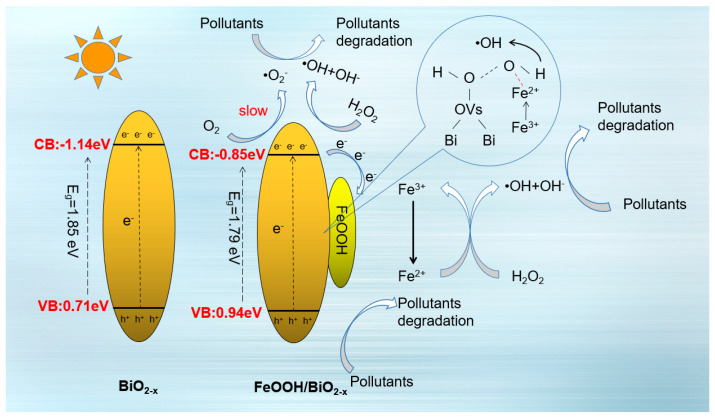
Degradation mechanism of the photo−Fenton system with 20FBO.

**Figure 11 molecules-29-00919-f011:**
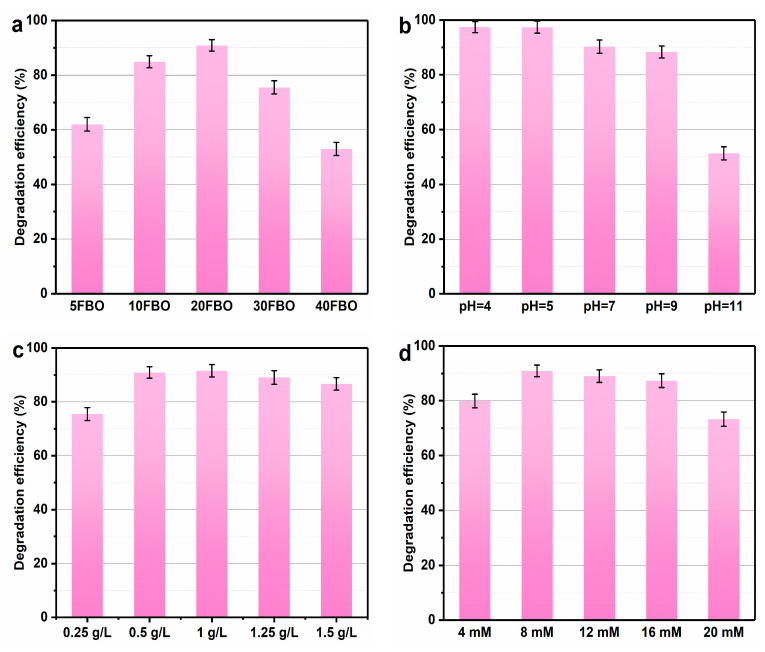
Photo-Fenton synergy condition optimization of 20BFO: (**a**) different loading ratios, (**b**) pH, (**d**) catalyst dosage, and (**c**) H_2_O_2_ concentration (conditions: MB concentration: 50 mg/L, catalyst concentration: 0.5 g/L, H_2_O_2_ concentration: 8 mM, initial pH = 7, illuminant source: 500 W Xe-lamp (λ > 420 nm), temperature: 30 ± 2 °C).

**Figure 12 molecules-29-00919-f012:**
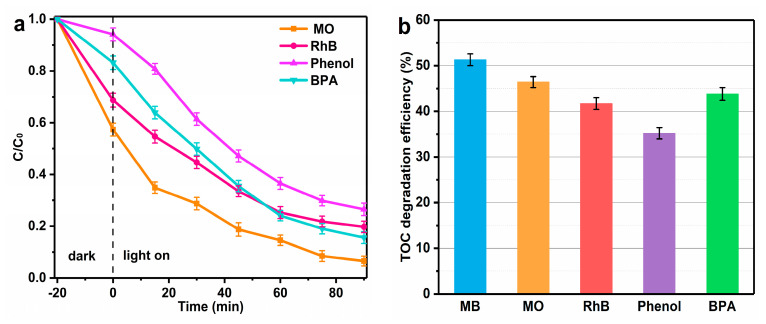
(**a**) Photo−Fenton degradation efficiency for different pollutants; (**b**) comparison of the TOC removal efficiencies of the photo-Fenton system within 90 min (C_0_ = 50 mg/L, Cat. = 0.5 g/L, H_2_O_2_ = 8 mM, pH = 7).

**Figure 13 molecules-29-00919-f013:**
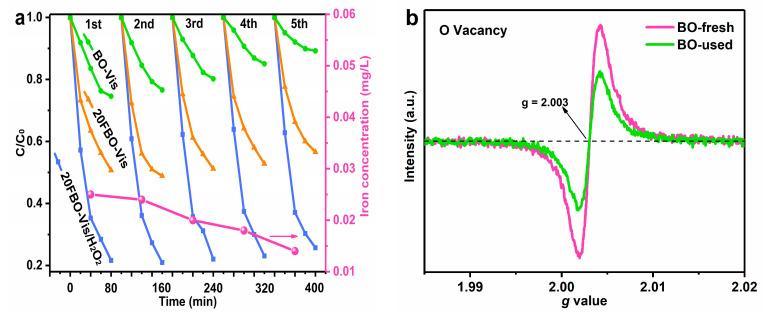
(**a**) Stability test of catalysts with different degradation systems under ultraviolet light and the curve of leached iron concentration; (**b**) comparison of ESR spectrum of BO catalyst before and after use.

## Data Availability

All data generated or analyzed during this study are included in this published article.

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
