# Peer review of "Broad Spectral Response FeOOH/BiO2−x Photocatalyst with Efficient Charge Transfer for Enhanced Photo-Fenton Synergistic Catalytic Activity"

_molecules, 2024, doi:10.3390/molecules29040919_

Round 1
Reviewer 1 Report
Comments and Suggestions for Authors
In this manuscript, Wu et al. reported on the synthesis of a binary composite photocatalyst by combining BiO2-X with amorphous FeOOH. Their primary goal is to address the issue of photocorrosion while simultaneously enhancing the catalytic activity of BiO2-X for the degradation of organic pollutants within the photo-Fenton system. The manuscript is well-written, presenting detailed and insightful analyses. With some minor revisions, it could be considered for publication. The specific points for revision are outlined below.
1. The abstract should focus on conveying the essential findings rather than providing an exhaustive account of all analyses. Additionally, ensure that abbreviations such as MB, FBO, and BO are defined upon their first use, and avoid unnecessary abbreviations like PPCPs, which only appear once.
2. Introduction: It is recommended to incorporate information that underscores the advantages of photo-induced technology over traditional non-light methods. See the provided reference (DOI: 10.1016/j.jphotochemrev.2023.100649).
3. The authors generated FBO using varying molar ratios of FeOOH, yet the selection of 20FBO as the optimized ratio lacks supporting data or explanation.
4. How was the EIS measurement performed? At open-circuit condition or at which applied potential?
5. Please also specify the intensity of the incident light used in the experiments.
6. Figure 5’s caption “Xe-lamp (λ > 420 nm, > 780 nm). I assume the authors meant (420 < λ <780 nm). Please double-check for clarity.
7. While it is evident that FeOOH played a crucial role in improving activity, the nearly seven-fold increase in surface area resulting from FeOOH decoration raises the question of whether the enhanced performance is solely attributed to the heightened number of active sites rather than intrinsic activity. A more in-depth discussion is needed.
Comments on the Quality of English Language
Minor editing of English language is needed, especially typos, abbreviation and capitalization.
Reviewer 2 Report
Comments and Suggestions for Authors
The researchers developed a promising composite material, FeOOH/BiO2-x, using a facile and eco-friendly synthesis method. The material exhibited high photodegradation activity under visible and near-infrared irradiation when used with hydrogen peroxide in a photo-Fenton process. Under optimal conditions, the material efficiently degraded various organic pollutants, demonstrating promise as an iron-based photo-Fenton catalyst. This is a comprehensive and well-executed study. The photocatalysts are characterized by an array of physicochemical techniques (TEM, XPS, ESR, DRS, BET, etc.). Thus, the article presents relevant research that can be recommended for publication in Molecules.
To further improve the manuscript, the authors should address the following minor points:
-In the experimental section, specify whether anhydrous or aqueous FeCl3 was used.
-The authors should specify the initial wavelength range used for photocatalysis irradiation. While a 420 nm cutoff filter was mentioned, the starting wavelength from the xenon lamp was not provided. Supplementing the xenon lamp parameters would clarify the irradiation conditions.
-Check and add page numbers for references 51, 54, 62.
-Figure captions in supplementary materials should be formatted.
-Do the resulting materials exhibit magnetic properties?
Reviewer 3 Report
Comments and Suggestions for Authors
The manuscript “Broad spectral response FeOOH/BiO2-x photocatalyst with efficient charge transfer for enhanced photo-Fenton synergistic catalytic activity” describes the preparation, characterization of novel catalysts and their application in photocatalytic degradation of dyes.
Abstract:
“This outstanding catalytic system offers a fresh approach to the creation and processing of iron-based photo-Fenton catalysts by quickly and efficiently degrading various organic contaminants.” In my opinion the time and degradation % should be mentioned here to support the statement.
Results and discussion:
I assumed that in according to “Molecules” rules for the authors the preparative section should be placed at the end of the manuscript. The authors stared the characterisation of the prepared samples by XRD, and it is OK, but I am confused with the abbreviation of samples. In my opinion some comments for example: the composites with different FeOOH:BiO ratio were prepared and give the example of the authors “logic” of abbreviation. It in my opinion it will facilitate the reading of the manuscript. In the text we have FeOOH, BO, FBO and in the figure we have BO, 5FBO, 10FBO, 20FBO and FeOOH it is very difficult to follow. That diffraction lines correspond to BiO3 (pyramidal) and BiO6 (octahedral)? Could the authors explain this in more details?
FTIR:
Why the authors use term “peak” during discussion of FTIR spectra? For spectroscopies (FTIR, Raman, UV-Vis etc.) always bands should be discussed not peaks. This has to be corrected in all the text. Here I have the same abbreviation problem as for XRD. Fig 1 a and b, why the sequence of XRD patterns and FTIR spectra is different? The authors maintain the line colours for each sample which is great, but the sequence of the spectra is different.
Without the explanation the of abbreviation “logic” the statement “The characteristic peaks of the FBO closely resemble those of pure BO, owing to the small amount of added FeOOH. Notably, at a mass percentage of 10% FeCl3, the characteristic peak exhibits the highest intensity.” has no sense.
The authors stated “SEM, TEM, and EDS (mapping) techniques were employed to investigate the structure and elemental composition of BO and FBO” Could the authors explain why and how?
Nitrogen adsorption-desorption was studied for BO and 20FBO samples, why not for other samples? The same question for XPS.
Could the authors explain the calculation of valent and conductive bands by XPS?
How the authors prove the MB degradation, by UV-VIS? There is a lack of MB adsorption experiments. The authors did not convince me that the MB degradation occurred under their experimental conditions, there is no analysis of product(s) formed. Why BO and 20FBO only were studies as (photo)catalysts? What was the pH of the reaction medium?
Fig 5 c, “ 2D HPLC chromatograms for photo-Fenton MB degradation”, Is it correct?
“The arc radius of 20FBO is significantly smaller than that of BO, indicating that its photogenerated charge experiences less resistance during transportation between interfaces.” I cannot see any arc…. in Fig 7b
I do not understand the discussion of EPR experiments DMPO-radicals. Could the authors explain? This part should be re-written, form the message of the authors (manuscript text) the reader (at least me) do not know how the radicals OH and O2 were “separated”. Only from the Figs. 8 a and 8 b legend I think that I started to understand how the experiment was performed. But this should be explained in the main text.
The model reaction (MB photo-degaradation) is not new and comparison with literature data is needed.
Summarising, I can support the manuscript for publication after proper corrections.
Round 2
Reviewer 3 Report
Comments and Suggestions for Authors
I support the manuscript to publish.